# The Biological Effect of Small Extracellular Vesicles on Colorectal Cancer Metastasis

**DOI:** 10.3390/cells11244071

**Published:** 2022-12-15

**Authors:** Xiaoxing Wang, Defa Huang, Jiyang Wu, Zhengzhe Li, Xiaomei Yi, Tianyu Zhong

**Affiliations:** 1The First School of Clinical Medicine, Gannan Medical University, Ganzhou 341000, China; 2Laboratory Medicine, First Affiliated Hospital of Gannan Medical University, Ganzhou 341000, China

**Keywords:** colorectal cancer, small extracellular vesicles, tumor microenvironment, angiogenesis, metastasis

## Abstract

Colorectal cancer (CRC) is a malignancy that seriously threatens human health, and metastasis from CRC is a major cause of death and poor prognosis for patients. Studying the potential mechanisms of small extracellular vesicles (sEVs) in tumor development may provide new options for early and effective diagnosis and treatment of CRC metastasis. In this review, we systematically describe how sEVs mediate epithelial mesenchymal transition (EMT), reconfigure the tumor microenvironment (TME), modulate the immune system, and alter vascular permeability and angiogenesis to promote CRC metastasis. We also discuss the current difficulties in studying sEVs and propose new ideas.

## 1. Introduction

Colorectal cancer (CRC) is a malignancy of the gastrointestinal tract with a high incidence and mortality rate. Recent statistics show that approximately 9,300,000 new cases of CRC occur worldwide each year, and that it has the second highest mortality rate (9.4%) of all cancers [1]. Metastasis is the spread of cancer cells to surrounding or distant organs and is a sign of poor prognosis. During this process, the normal colorectal mucosal epithelium is transformed to hyperproliferative epithelial cells by multiple factors, leading to the development of CRC. When they lose their normal tissue structure, hyperproliferative intestinal epithelial cells (IECs) form adenomas which invade the underlying mucosa and surrounding tissues to become cancerous [2]. Metastasis is an inefficient process because most cancer cells do not acquire the necessary capacity to regenerate tumors at distant sites. To be able to metastasize, cancer cells must invade surrounding tissues, survive in circulation, colonize distant organs and eventually grow in other organs [3]. In addition to direct cell–cell contact, abnormally proliferating tumor cells promote tumor growth and development by releasing large amounts of substances that cause epithelial mesenchymal transition (EMT) of the intestinal epithelium and colonization of distant organs via the circulation [4]. EMT has been proved to play a key role in the metastasis cascade of epithelial carcinoma. The tumor microenvironment (TME) promotes cancer cell proliferation and metastasis by recruiting immunosuppressive cells, remodeling the extracellular matrix, and promoting angiogenesis [5]. In addition, tumor-associated macrophages (TAM) promote cancer development, progression and metastasis through intercellular communication with cancer cells [6]. Increased vascular permeability and angiogenesis are the main bases for tumor growth and metastasis [7]. These factors interact and ultimately promote CRC metastasis. The five-year survival rate for regional CRC is approximately 72%; however, metastasis reduces the survival rate to 15% [8]. Metastasis is a key factor affecting CRC treatment and prognosis, and patients who develop metastasis have limited treatment options and a poor prognosis [9]. Exploring the mechanisms by which metastasis occurs in CRC may provide new treatment options for patients with metastatic colorectal cancer.

Small extracellular vesicles (sEVs) are the major component of extracellular vesicles (EVs), which range in size from 30 to 150 nm. sEVs originate from endosomes and are formed by the fusion of multivesicular bodies (MVBs) with the plasma membrane. MVBs contain intraluminal vesicles that are released into the extracellular lumen when these MVBs fuse with the plasma membrane [10,11]. The biogenesis of EVs is regulated primarily through endosomal sorting complexes required for transport (ESCRT) or lipid ceramides [12]. The surface and interior of sEVs membranes are rich in proteins and consist mainly of nonspecific and specific proteins. Non-specific proteins include members of the four transmembrane protein family (CD9, CD63, CD81), membrane fusion and transport-related proteins such as Alix, TSG101, and cytoskeletal proteins such as microtubulin and troponin. These not only maintain the stability of the structure and function of sEVs, but also act as surface protein markers in the sorting and identification of sEVs [13,14]. The specific proteins are a class of sEV proteins involved in the physiologic and pathologic processes of the organism, and they play an important role in the development of specific diseases [15]. During sEV formation and secretion, lipids, proteins, nucleic acids, and other components are carried from the donor cell to the recipient cell [16]. sEVs can transport biomolecules from tissues to body fluids [17] and through body fluids to distant tissues and organs. These properties contribute to the role of sEVs in intercellular communication, where signaling molecules shuttle between nearby and remote cells, increasing the invasive capacity of tumor cells and promoting tumor metastasis. There is growing evidence that sEVs are critical factors mediating cancer metastasis [18,19]. Through sEVs, highly metastatic tumor cells can transfer biomolecules to less malignant cells, which then may begin to display enhanced migratory and metastatic behavior.

Several studies have shown that differential expression of sEV content is closely associated with CRC metastasis and plays an important role in the multi-step and multi-linked metastatic process [20,21,22]. Tumor-associated sEVs promote tumor cell metastasis and xenotropic colonization through various mechanisms such as influencing tumor cells to undergo EMT, suppressing tumor immunity, promoting generation of the pre-metastatic ecological niche, and increasing vascular leakage [23,24,25]. Other mechanisms are summarized in Figure 1. Compared with healthy individuals, sEVs are more abundantly expressed in the circulation, bodily fluids, and local tissues of cancer patients, providing an important biological basis for tumorigenesis and progression [26]. Therefore, this paper highlights evidence showing that sEVs affect CRC metastasis by mediating EMT, remodeling the TME, macrophage M2 polarization, and increasing vascular leakage and angiogenesis. We also summarize the potential mechanisms of sEVs in CRC development to provide new ideas for early and effective diagnosis and treatment options for CRC metastases.

## 2. Small Extracellular Vesicles Affect Colorectal Cancer Metastasis by Remodeling the Tumor Microenvironment

CRC cells undergo a series of complex and highly regulated processes as they metastasize to surrounding and distant organs and tissues. The TME plays a crucial role in the process of tumor metastasis. The TME refers to the surrounding microenvironment where tumor cells exist, and includes surrounding blood vessels, immune cells, fibroblasts, inflammatory cells, various signaling molecules, and the extracellular matrix (ECM) [27]. The fundamental role of TME is to interact dynamically with malignant tumor cells [28]. TME promotes tumor cell proliferation and metastasis by recruiting immunosuppressive cells, remodeling the ECM, and promoting angiogenesis [5]. sEVs are currently thought to regulate the TME by playing a key signaling mediator role (Table 1). Tumor cells secrete sEVs that act on various cells in the TME, including fibroblasts, endothelial cells, and local immune cells, to induce a pro-tumor phenotype in different cells. Then, the tumor-affected cells release sEVs, which in turn increase the motility and invasive potential of tumor cells, promote angiogenesis and the formation of pre-metastatic ecological sites, and facilitate drug resistance. Through these and other mechanisms, sEVs promote the tumor metastasis cascade [29].

It was found that sEVs derived from highly metastatic CRC cells enriched in miR-181a-5p-activated hepatic stellate cells (HSCs) by activating the IL6/STAT3 signaling pathway, acting directly on suppressor of cytokine signaling 3 (SOCS3) and causing TME remodeling. Activated HSCs release C-C motif chemokine ligand 20 (CCL20), which upregulates miR-181a-5p expression in CRC via the ERK/Elk-1 pathway. In addition, sEVs enable a positive feedback loop between CRC cells and HSCs, ultimately leading to CRC liver metastasis (CRLM) [30]. Exposure to sEVs from highly metastatic CRC cells was shown to increase the invasive and metastatic capacity of primary CRC cells, and both in vivo and in vitro experiments have demonstrated that sEVs derived from highly metastatic CRC cells mediated hypo-metastasis through metastases-associated lung adenocarcinoma transcript 1 (MALAT1). MALAT1 mediates fucosyltransferase 4 (FUT4)-related glycosylation in hypometastatic colorectal cancer cells and activates the PI3K/AKT/mTOR pathway, affecting the tumor microenvironment and allowing CRC to metastasize [31].

CRC cells originate from sEVs that highly express circPACRGL. CircPACRGL has been shown to enhance CRC cell proliferation, migration, and invasion via the miR-142-3p/miR-506-3p-TGF-β1 axis. It also modulates N1-N2 neutrophil differentiation and regulates the differentiation of tumor-associated neutrophils (TANs) in the TME, allowing immunosuppression to occur and providing an important basis for CRC metastasis [32]. CRC cell-derived HSPC111-containing sEVs alter lipid metabolism by increasing the levels of acetyl coenzyme A in cancer-associated fibroblasts (CAFs), ultimately promoting pre-metastatic niche formation and CRLM. This demonstrates that CRC-derived sEVs are key mediators in establishing the pre-metastatic niche to promote CRC cell liver metastasis [33]. The CRC-derived sEVs deliver miR-221/222, which plays an important role in promoting the liver-specific metastasis of CRC by remodeling the hepatic microenvironment. sEVs secreted by CRC transport miR-221/222 to hepatic stromal cells and activate hepatocyte growth factor (HGF) by inhibiting serine peptidase inhibitor and Kunitz type 1 (SPINT1) expression. This induces the formation of a suitable metastatic and colonization environment for incoming metastatic tumor cells, thus contributing to CRC metastasis [34]. CRC cell-derived miR-146a-5p and miR-155-5p can be taken up by cancer-associated fibroblasts (CAFs) via sEVs and act through the JAK2-STAT3/NF-KB signaling pathway to suppress suppressor of cytokine signaling 1 (SOCS1) and zinc finger and BTB domain-containing 2 protein (ZBTB2). This allows activated CAFs to further potentiate the invasive capability of CRC cells. Mechanistic studies revealed that levels of inflammation-related cytokines IL-6, tumor necrosis factor-α, transforming growth factor, and C-X-C motif chemokine ligand 12 (CXCL12) increased significantly in CAFs transfected with miR-146a-5p and miR-155-5p. This triggered mesenchymal transformation and pre-metastatic conversion of CRC cells, ultimately causing CRC lung metastasis. These data were further confirmed in animal studies [35]. SEVs derived from CRC cells overexpressing cationic amino acid transporter 1 (CAT1) significantly enhanced the growth of vascular endothelial cells and tubular formation, promoted angiogenesis, and accelerated CRC metastasis by regulating cGMP metabolism and up-regulating arginine transport and downstream nitric oxide metabolism [36]. The formation of pre-metastatic ecotone is an important factor involved in tumor metastasis. It was shown that CRC cells release integrin beta-like 1 (ITGBL1)-rich sEVs into the circulation by stimulating tumor necrosis factor (TNF) alpha-induced protein 3 (TNFAIP3)-mediated nuclear factor kappa-B (NF-κB) signaling pathway activation in distant fibroblasts. The activated fibroblasts induced the formation of the pre-metastatic niche and promoted metastatic cancer growth by secreting pro-inflammatory cytokines (e.g, IL-6 and IL-8) [37]. MicroRNA-21-5p (miR-21) is highly expressed in colorectal cancer-derived sEVs, polarizes hepatic macrophages to an interleukin 6 (IL-6) producing phenotype via toll-like receptor 7 (TLR7), induces the formation of an inflammatory microenvironment, and promotes CRC liver-specific metastasis [38].

Hypoxic CRC cells secrete sEVs that promote the migration and invasion of normoxic CRC cells. These hypoxic CRC cell-derived sEVs were found to contain Wnt4, which affects hypoxic extracellular vesicle-mediated migration and invasion of normoxic CRC cells, enhances pro-metastatic behaviors, and further mediates CRC metastasis [39]. sEVs derived from CAF cell culture supernatants highly expressing circEIF3K mediate hypoxic CRC progression and metastasis via the circEIF3K/miR-214/PD-L1 axis [40]. To further demonstrate the effect of sEVs on CRC metastasis in the tumor microenvironment, Qu et al. conducted in vivo experiments and found that knockdown of circN4BP2L2 in CAF-derived sEVs inhibited subcutaneous tumorigenesis and liver metastasis in CRC nude mice [41].

## 3. Small Extracellular Vesicles Mediate Epithelial Mesenchymal Transition Affecting Colorectal Cancer Metastasis

Epithelial mesenchymal transition (EMT) plays a key role in tumorigenesis and metastasis. During EMT, epithelial tumor cells undergo marked morphological and phenotypic changes, including loss of tight junctions, cell polarity, and cytoskeletal reorganization, including a more aggressive phenotype [42]. In addition, tight junction proteins regulate para-cellular permeability and maintain cell polarity [43]. The absence of tight junction proteins can lead to the development of EMT and further promote cancer progression [44,45]. Evidence suggests that the expression of Claudin-2 in small extracellular vesicles of patient blood origin can be a relevant prognostic biomarker for predicting the development of replacement type liver metastases in patients with colorectal cancer [46]. EMT is thought to be a prerequisite for causing initial tumor cells to become motile and aggressive, leading to metastasis and recurrence of many cancer types [47]. Many studies have demonstrated that during the progression of many types of cancer such as lung, bladder, and liver cancer, tumor-associated sEVs carry and release a variety of active components such as miRNAs, proteins, and circRNAs into the surrounding cells. This further promotes the metastasis of tumor cells by causing EMT of the recipient cells through various pathways [24,48,49]. In addition, sEVs affect the EMT of recipient cells and play an important role in CRC metastasis (Table 2).

Cancer-associated fibroblasts (CAFs) are key stromal cells that play a dominant role in tumor progression. Researchers have co-cultured sEVs derived from CAFs with CRC cells. CRC cells internalized sEVs containing miR-92a-3p, leading to a significant increase in the level of miR-92a-3p in CRC cells. In addition, sEVs from CAFs with high miR-92a-3p expression activated the Wnt/βcatenin pathway in CRC cells, inhibiting mitochondrial apoptosis by directly inhibiting FBXW7 and MOAP1 expression and promoting EMT in CRC cells, thus further facilitating metastasis [50].

Another study showed that sEVs derived from metastatic SW620 CRC cells targeted RAS p21 protein activator 1 (RASA1) via miR-335-5p. Downregulation of RASA1 in CRC cells activated signaling downstream of RAS and induced EMT, further promoting CRC cell invasion and metastasis [51]. High levels of plasmacytoma variant translocation 1 (PVT1) in serum-derived sEVs from patients with distant CRC metastases, as well as treatment of CRC cells with sEVs expressing PVT1, increased intracellular expression of EGFR and VEGFA and promoted EMT [52]. Additionally, high expression of miR-106b-3p in serum-derived sEVs from CRC patients with distant metastases led to downregulation of deleted in liver cancer 1 (DLC-1) in tumor cells and ultimately inhibited CRC metastasis [53]. HCT-8 colorectal cancer cells actively secreted miR-210-containing sEVs to promote the expression of key EMT proteins (E-cadherin-negative and vimentin-positive). Additionally, miR-210-containing sEVs have been found to promote EMT, thereby promoting CRC metastasis [54].

Studies have shown that co-culture of LO2 hepatocytes and phosphatase of regenerating liver-3 (PRL-3) overexpressing CRC cells induced the re-expression of E-cadherin in CRC cells. The re-expression of E-cadherin is an important marker for the development of EMT in CRC. In addition, Src plays an important role in regulating EMT through activation of epidermal growth factor receptor (EGFR) [55]. The above evidence suggests that hepatocytes can mediate the roles of E-cadherin and Src in the epithelial mesenchymal transition process of colorectal carcinogenesis, providing a new potential mechanism of colorectal cancer liver metastasis. To investigate the mechanism by which hepatocytes affect the development of liver metastasis in colorectal carcinoma, Xu et al. found that sEVs from hepatocytes inhibited Src expression and EGFR activation in CRC cells via miR-203a-3p and promoted E-cadherin re-expression, thereby inducing EMT. These related studies reveal a mechanism of liver metastasis in CRC cells and provide a comprehensive understanding of CRC liver metastasis [56].

High expression of integrin alpha6 (ITGA6) activated TGF-1 in CRC cells to promote CRC progression and metastasis. MiR-3940-5p in mesenchymal stem cell-derived sEVs promoted the progression and metastasis of CRC by downregulating ITGA6 in CRC cells. Reduced ITGA6 levels disrupted the TGF-β1 signaling pathway, inhibiting CRC cell invasion and EMT, and ultimately reducing tumor growth and metastatic capacity [57]. HCT116 cells were induced to undergo EMT following treatment with IL-6, and purified HCT116 cell supernatant-derived sEVs were found to overexpress miR-128-3p in the vesicles. MiR-128-3p overexpressed sEVs were subsequently transferred to normal HCT-116 target cells, which induced EMT via TGF-β/SMAD and JAK/STAT signaling and further promoted the development of distant CRC metastasis [58].

## 4. Small Extracellular Vesicles Promote Macrophages to Undergo M2 Polarization Affecting Colorectal Cancer Metastasis

Tumor-associated macrophages are derived from circulating peripheral blood monocytes, which are derived from the bone marrow. These monocytes are recruited into tumor tissues and undergo local differentiation in response to various cytokines, chemokines and growth factors produced by stromal and tumor cells in the TME, affecting tumor progression [59]. TAMs have two opposing phenotypes: M2 subtype macrophages exhibit pro-tumor activity and M1 subtype cells have anti-tumor activity. When CRC cells interact with M2 subtype TAMs, M2 TAMs can induce EMT in tumor cells by secreting IL-6, which promotes CRC metastasis. In addition, EMT-programmed CRC cells can activate the expression of different cytokines (e.g, CCL2 and IL-4) in cancer cells to enhance macrophage recruitment and promote M2 polarization [60,61]. sEVs secreted by tumor cells are taken up by macrophages and differentiated into tumor-associated macrophages of the M2 phenotype, which promotes tumor growth and immunosuppression [62]. Here, we summarize the relevant studies showing that sEVs promote CRC metastasis through M2 polarization of macrophages (Table 3).

Tumor-derived sEVs high in miR-934 can promote CRLM by regulating the interaction between CRC cells and TAMs. Studies have shown that tumor–TAM interactions in the metastatic microenvironment are mediated by tumor-derived sEVs and affect CRLM. Phosphatase and tensin homolog (PTEN) expression and activation of the PI3K/AKT signaling pathway induce M2 macrophage polarization. Polarized M2 macrophages can induce pre-metastatic niche formation and promote CRLM by secreting C-X-C motif chemokine ligand 13 (CXCL13) [63]. CRC cell-derived sEVs translocate lncRNARPPH1 into macrophages and mediate their M2 polarization, thereby promoting CRC cell metastasis and proliferation [64]. C-X-C motif chemokine receptor 4 (CXCR4)-overexpressing CRC cells deliver multiple miRNAs (miR-25-3p, miR-130b-3p and miR-425-5p) to macrophages via sEVs, causing M2 polarization of macrophages via the PTEN/PI3K/Akt signaling pathway. This promotes EMT and secretion of vascular endothelial growth factor (VEGF), enhancing CRC metastasis [65]. In addition, one study found that lncRNA HLA-F-AS1 promoted profilin 1 (PFN1) expression in CRC-sEVs by suppressing miR-375, thereby polarizing macrophages toward the M2 phenotype and exacerbating CRC tumorigenesis and metastasis [66]. CRC-sEVs activated macrophage nucleotide-binding oligomerization domain 1 (NOD1) signaling and promoted secretion of the pro-inflammatory cytokines IL-6 and tumor necrosis factor-α (TNF-α). NOD1-activated macrophages also promoted CRC cell metastasis [67]. Takano et al. found that sEVs carrying miR-203 from CRC cells were integrated into monocytes and promoted the expression of M2 markers. Studies have also demonstrated that miR-203 promoted the differentiation of monocytes to M2 TAMs. In a xenotransplantation mouse model, more liver metastases occurred in mice bearing CRC cells transfected with miR-203 compared to controls. Additionally, ex vivo experiments elucidated that sEVs of CRC cell origin promoted M2 polarization of macrophages via miR-203, laying the foundation for CRC liver metastasis [68]. Similarly, Wang et al. found that M2-type macrophage-derived sEVs transported miR-21-5p and miR-155-5p into CRC cells, allowing miRNA targeting to the BRG1 coding sequence, downregulating BRG1 expression and promoting CRC metastasis [69].

In addition to CRC cell-derived sEVs, CRC cells with EMT function can also promote CRC metastasis by causing M2 polarization in macrophages. It was found that EMT-CRC cell-derived extracellular vesicles highly expressing microRNA-106b-5p promoted M2 polarization of macrophages by directly inhibiting activation of the phosphatidylinositol 3-kinase (PI3K) γ/AKT/mTOR signaling pathway at the post-transcriptional level of programmed cell death 4 (PDCD4). Activated M2 macrophages promoted EMT-mediated migration, invasion and metastasis of CRC cells in a positive feedback manner [70]. In addition, TP53 mutants (mutp53) have been implicated in the pathogenesis of most human cancers. Specific gain-of-function (GOF) mutp53 proteins do not exhibit the tumor suppressor activity of the wild-type proteins. Colon cancer cells carrying GOF mutp53 selectively shed miR-1246-rich sEVs and promote macrophage M2 polarization to promote colorectal cancer metastasis [71].

## 5. Small Extracellular Vesicles Increase Vascular Leakage and Angiogenesis to Promote Colorectal Cancer Metastasis

Tumor growth, invasion, and metastasis depend on angiogenesis for adequate oxygen and nutrient supply to the tumor cells. Moreover, disruption of vascular integrity and the consequent increase in vascular permeability allows EMT tumor cells to enter the vasculature and subsequently metastasize. sEVs secreted by cancer cells have been reported to be taken up by neighboring or distant receptor cells to promote tumor development and metastasis. They play an important role in tumor angiogenesis and vascular leakage [72,73,74]. Table 4 summarizes mechanistic studies showing that sEVs affect CRC metastasis by increasing vascular leakage and angiogenesis.

Angiopoietin-like protein 1 (ANGPTL1) protein levels were found to be significantly downregulated in sEVs from CRC tissues compared to paired normal tissues. sEVs with downregulated ANGPTL1 decreased matrix metallopeptidase 9 (MMP9) levels in Kupffer cells (KCs) by inhibiting the JAK2-STAT3 signaling pathway, preventing hepatic vascular leakage and ultimately inhibiting the development of liver metastasis in CRC [75]. Zeng et al. found that CRC cell-derived sEVs carrying miR-25-3p targeted Krüppel-like factor 2 (KLF2) and Krüppel-like factor 4 (KLF4) to regulate the expression of VEGFR2, ZO-1, occludin, and claudin-5 in endothelial cells, thereby promoting vascular permeability and angiogenesis. In vivo assays demonstrated that miR-25-3p-containing sEVs significantly induced vascular leakage and enhanced CRC metastasis in mouse liver and lung tissues. Furthermore, the expression level of miR-25-3p in circulating sEVs was significantly higher in CRC patients with metastasis than in those without metastasis. This suggests that sEVs with high miR-25-3p expression can ultimately lead to distant CRC metastasis by increasing vascular permeability and promoting angiogenesis [76]. It has been reported that CRC-sEVs target suppressor of cytokine signaling 3 (SOCS3) in endothelial cells via miR-221-3 to regulate the STAT3/VEGFR-2 signaling axis. This leads to promotion of endothelial cell proliferation, migration, and formation of vascular-like structures, ultimately affecting CRC metastasis [77]. It was demonstrated that early growth response-1 (Egr-1) activation in endothelial cells is associated with the angiogenic activity of CRC cell-derived sEVs. Activation of Egr-1 by CRC cell-derived sEVs was shown to promote endothelial cell migration and induce angiogenesis via the ERK1/2 and JNK signaling pathways [78].

In addition, researchers found that sEVs secreted by EMT-CRC cells further disrupted endothelial cell junctions by transferring miR-27b-3p into vascular endothelial cells and attenuating vascular endothelial cadherin (VE-Cad) and p120-catenin (p120) expression in a post-transcriptional manner. This negatively affected cell integrity, increased vascular permeability and ultimately induced cancer metastasis [79]. The adenomatous polyposis coli (APC) gene plays a key role in the pathogenesis of CRC. Xie et al. found that sEVs from CRC cells with lncRNA-APC1 silencing promoted angiogenesis and CRC metastasis by activating the mitogen-activated protein kinase (MAPK) pathway in vascular endothelial cells [80].

## 6. Discussion

Because sEVs are formed by the fusion of multivesicular bodies (MVBs) with the plasma membrane, they serve as important carriers of intercellular information and participate in a variety of physiological activities of the body, playing a crucial role in pathological states. Most studies have focused on the role of sEVs in the development of cancer, diagnosis, drug delivery, and prognosis. During the development of malignant tumors, sEVs play a role in the regulation of the local microenvironment by carrying and transporting tumor cell content, tumor-associated fibroblasts, and tumor-associated immune cells, which are important to the regulation of tumor development in target cells. They release their content after binding to target cells, causing an intracellular cascade reaction. Cancer metastasis is a process involving the spread of cancer cells from primary lesions to distant organs and is a major cause of cancer lethality. Numerous studies have demonstrated that sEVs contribute to tumor metastasis by promoting EMT of tumor cells, inducing angiogenesis, promoting vascular leakage, establishing a pre-metastatic microenvironment, immunomodulation and forming a pre-metastatic ecological niche during cancer metastasis [81,82]. Being involved in every step of the tumor cell metastasis process, sEVs provide a new direction for the study of cancer metastasis and clinical translational research. sEVs play an important role in the development and progression of CRC, and many investigators have conducted exploratory studies on sEVs as important components of the TME in the context of early diagnosis and treatment of CRC. However, the specific mechanisms involved in tumor progression remain unclear [83,84]. Many questions regarding the mechanisms of tumor-derived extracellular vesicles in CRC metastasis remain. In addition, the sensitivity and specificity of sEV use in the diagnosis and treatment of colorectal cancer is sub-optimal. Existing techniques for the isolation of small extracellular vesicles, such as ultracentrifugation and size exclusion, fail to achieve ideal purity and yield. In addition, characterization methods and storage stability are also important issues in the clinical application of sEVs [85,86]. Further, due to the heterogeneity of sEVs, specific tools for differentiating sEVs from different intracellular origins are under development. This is needed to properly assess the molecular mechanisms of biogenesis and secretion as well as the respective functions of EV subtypes [87].

Regarding the function of sEVs in promoting cancer metastasis through immunomodulation, previous researchers have mainly focused on cancer-derived sEVs that exert tumor–host immunosuppressive functions and promote tumor progression by binding to and altering the biological functions of surface receptors on natural killer cells, dendritic cells, T and B lymphocytes, and mast cells [88,89,90]. Recently, studies have focused on the mechanisms through which sEVs induce differentiated macrophages to form M2-type tumor-associated macrophages, providing new evidence for immunotherapy in cancer [91]. 

We have summarized mechanistic studies showing how sEVs induce macrophages to undergo M2 polarization, and thus disrupt anti-tumor immunity and enable CRC to metastasize. We have also discussed sEV-induced inhibition of M2 polarization to reduce colorectal cancer metastasis. Therefore, sEVs are a promising new target that may be used to reduce colorectal cancer metastasis. This will be the focus of our future research.

In this review, we provide a detailed summary of how sEVs mediate the epithelial-mesenchymal transition, reconfigure the tumor microenvironment, modulate the immune system, and alter vascular permeability and angiogenesis to promote CRC metastasis. Importantly, these studies have identified several sEVs components such as microRNA, circRNA, lncRNA, and proteins highly associated with CRC metastasis. This work improves our understanding of the role of sEVs in the development of CRC and provides a basis for subsequent studies. Greater attention to the mechanisms of small extracellular vesicles in tumor progression and efforts in translational medicine research will provide a richer foundation for the clinical role of sEVs in the early diagnosis and treatment of colorectal cancer.

In conclusion, sEVs play an important role in CRC metastasis, and an in-depth exploration of the function of sEVs in metastasis will help identify early therapeutic approaches for CRC. Through summarizing the currently available studies, we have increased the understanding of how sEVs promote metastasis and organotropism. The translation of this knowledge is very useful for clinical cancer treatment. Future studies should focus on examining whether sEVs from metastatic CRC sites facilitate further metastatic tumor formation in other distant organs. Seeking answers to these questions will not only increase our understanding of sEVs in CRC biology but also provide important evidence for much-needed novel therapies for advanced CRC. Furthermore, increasing the number of studies focused on sEVs and CRC metastasis will facilitate the development of sEVs as a biomarker for metastatic CRC.

## Figures and Tables

**Figure 1 cells-11-04071-f001:**
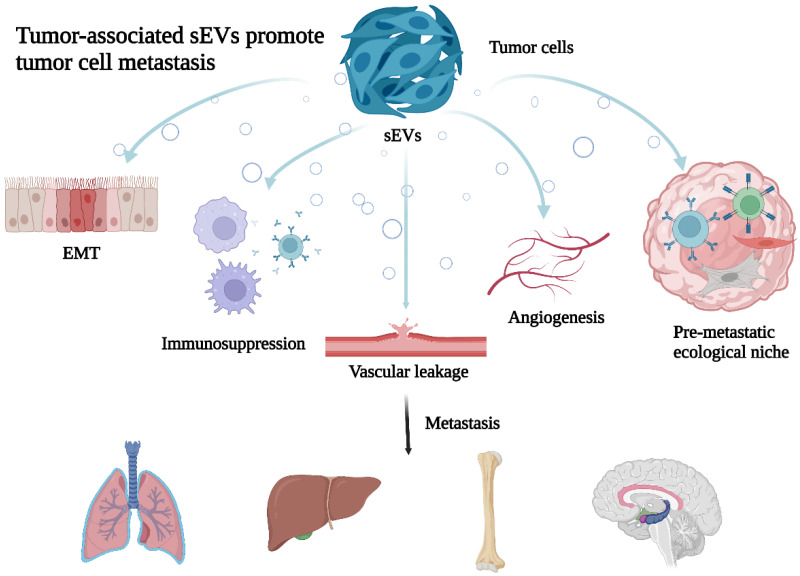
Tumor-associated sEVs promote tumor cell metastasis. Tumor-derived sEVs through promoting cellular epithelial–mesenchymal transformation (EMT), reshaping the tumor microenvironment (TME), immunosuppression, increase vascular leakage and angiogenesis to promote cancer metastasis.

**Table 1 cells-11-04071-t001:** sEVs affect CRC metastasis by remodeling the TME.

Cancer Type	sEVs Cargos	Tissues and/or Cells	Mechanism	Function	Refs
Colorectal cancer	miR-181a-5p	HT29, SW480, RKO, SW620 and plasma from CRLM	Promote liver metastasis by activating hepatic stellate cells and remodeling the tumor microenvironment	Promote metastasis	[30]
Colorectal cancer	MALAT1	LoVo, HCT-8, SW620, SW480 and CRC tissues	Promote the malignant behavior of CRC cells by sponging miR-26a/26b via regulating FUT4 and activating PI3K/Akt/mTOR pathway	Promote metastasis	[31]
Colorectal cancer	circPACRGL	HCT116 and SW480	Enhance CRC cell proliferation, migration and invasion, as well as differentiation of N1-N2 neutrophils via miR-142-3p/miR-506-3p-TGF-β1 axis	Promote metastasis	[32]
Colorectal cancer	HSPC111	HCT116, SW620, HT29 and SW480	Facilitate pre-metastatic niche formation and CRLM	Promote metastasis	[33]
Colorectal cancer	miR-221/222	Tissues and SW480	Induce the formation of a hospitable metastatic environment, providing an appropriate colonization environment for incoming metastatic tumor cells	Promote metastasis	[34]
Colorectal cancer	miR-146a-5p and miR-155-5p	HCT116 and SW620	Promote CXCL12/CXCR7-induced metastasis of colorectal cancer by crosstalk with cancer-associated fibroblasts	Promote metastasis	[35]
Colorectal cancer	CAT1	HCT116 and tissues	Enhance vascular endothelial cell growth and tubule formation via up-regulation of arginine transport and downstream NO metabolic pathway	Promote metastasis	[36]
Colorectal cancer	ITGBL1	Tissues and NCM460, SW480 and SW620	Promote metastatic cancer growth by secreting pro-inflammatory cytokine	Promote metastasis	[37]
Colorectal cancer	microRNA-21-5p	SW480, SW620 and LoVo	Promote liver metastasis by inducing an inflammatory premetastatic niche	Promote metastasis	[38]
Colorectal cancer	Wnt4	HT29 and HCT116	Enhance pro-metastatic behaviors	Promote metastasis	[39]
Colorectal cancer	circEIF3K	CAF	Promote CRC progression via miR-214/PD-L1 axis	Promote metastasis	[40]
Colorectal cancer	circN4BP2L2	CAF	Promote subcutaneous tumorigenesis and liver metastasis in CRC nude mice	Promote metastasis	[41]

**Abbreviations:** CRLM liver metastasis of colorectal cancer, CRC colorectal cancer, CAF cancer-associated fibroblast, MALAT1 metastases-associated lung adenocarcinoma transcript 1, FUT4 fucosyltransferase 4, CAT1 cationic amino acid transporter 1, CXCL12 C-X-C motif chemokine ligand 12, ITGBL1 integrin beta-like 1.

**Table 2 cells-11-04071-t002:** sEVs related with colorectal cancer metastasis by EMT.

Cancer Type	sEVs Cargos	Tissues and/or Cells	Mechanism	Function	Refs
Colorectal cancer	miR-92a-3p	CAFs and serum from CRC	Activate Wnt/β-catenin pathway contributing to cell stemness, EMT, metastasis and in CRC	Promote metastasis	[50]
Colorectal cancer	miR-335-5p	SW620	Promotes CRC invasion and metastasis by facilitating EMT via targeting RASA1	Promote metastasis	[51]
Colorectal cancer	PVT1	Serum from metastatic CRC	Increase expression of metastatic markers such as VEGFA and EGFR	Promote metastasis	[52]
Colorectal cancer	miR-106b-3p	Serum from metastatic CRC	Decrease the expression of DLC-1 and Inhibit the EMT	Inhibit metastasis	[53]
Colorectal cancer	miR-210	HCT-8	Promote the expression of key EMT proteins	Promote metastasis	[54]
Colorectal cancer	miR-203a-3p	LO2	Induce MET in PRL-3 overexpressing CRC cells	Promote metastasis	[56]
Colorectal cancer	microRNA-3940-5p	MSC	Inhibit colorectal cancer metastasis by Targeting Integrin α6	Inhibit metastasis	[57]
Colorectal cancer	miR-128-3p	HCT116 cell culture fluid and serum	Induce the activation of TGF-β/SMAD and JAK/STAT3 signaling in CRC cells and xenografted tumors, which led to EMT	Promote metastasis	[58]

**Abbreviations:** CAFs cancer-associated fibroblasts, CRC colorectal cancer, MSC mesenchymal stem cell, EGFR epidermal growth factor receptor, EMT epithelial mesenchymal transformation, PVT1 plasmacytoma variant translocation 1, PRL-3 regenerating liver-3.

**Table 3 cells-11-04071-t003:** sEVs promote macrophages to undergo M2 polarization affecting CRC metastasis.

Cancer Type	sEVs Cargos	Tissues and/or Cells	Mechanism	Function	Refs
Colorectal cancer	miR-934	FHC, SW480, SW620, HCT-8, HT-29, CaCo2, LoVo and RKO	Induce macrophage M2 polarization to promote liver metastasis of colorectal cancer	Promote metastasis	[63]
Colorectal cancer	LncRNA RPPH1	HCT8, SW620 and HT29	Promote colorectal cancer metastasis by interacting with TUBB3 and by promoting exosome-mediated macrophage M2 polarization	Promote metastasis	[64]
Colorectal cancer	miR-25-3p, miR-130b-3p and miR-425-5p	HCT116	Contribute to CXCL12/CXCR4-induced liver metastasis of colorectal cancer by enhancing M2 polarization of macrophages	Promote metastasis	[65]
Colorectal cancer	LncRNA HLA-F-AS1	DLD-1, HT-29, SW620, SW480,HCT116, and Caco-2	Promote colorectal cancer metastasis by inducing PFN1 in colorectal cancer-derived extracellular vesicles and mediating macrophage polarization	Promote metastasis	[66]
Colorectal cancer	-	HT-29, HCT116 and plasma	Promote metastasis via the NOD1 signaling pathway	Promote metastasis	[67]
Colorectal cancer	microRNA-203	Serum, tissues, CaR-1, RKO, Colo205, Colo320DM, DLD1, HCT116, Lovo, SW480 and SW620	Promote metastasis possibly via inducing tumor-associated macrophages in colorectal cancer	Promote metastasis	[68]
Colorectal cancer	miR-21-5p and miR-155-5p	SW48, SW480, and CO-115	Down-regulate BRG1 expression and promote CRC metastasis	Promote metastasis	[69]
Colorectal cancer	microRNA-106b-5p	HCT116 and HT29	Activate EMT-cancer cell and M2-subtype TAM interaction to facilitate CRC metastasis	Promote metastasis	[70]
Colorectal cancer	miR-1246	HT-29 and HCT116	Promote macrophage M2 polarization to promote colorectal cancer metastasis	Promote metastasis	[71]

**Abbreviations:** CRC colorectal cancer, CXCL13 C-X-C motif chemokine ligand 12, CXCR4 C-X-C motif chemokine receptor 4, NOD1 nucleotide-binding oligomerization domain 1.

**Table 4 cells-11-04071-t004:** sEVs increase vascular leakage and angiogenesis to promote CRC metastasis.

Cancer Type	sEVs Cargos	Tissues and/or Cells	Mechanism	Function	Refs
Colorectal cancer	ANGPTL1	SW620 and tissues	Attenuate CRLM by regulating Kupffer cell secretion pattern and impeding MMP9-induced vascular leakiness	Inhibit metastasis	[75]
Colorectal cancer	miR-25-3p	SW480, LS174T, SW620, LOVO, HCT116 and tissues	Promote pre-metastatic niche formation by inducing vascular permeability and angiogenesis	Promote metastasis	[76]
Colorectal cancer	miR-221-3p	HCT116 and Caco2	Promote endothelial cell angiogenesis via targeting suppressor of cytokine signaling 3	Promote metastasis	[77]
Colorectal cancer	-	SW480 and HCT116	Egr-1 activation by cancer-derived extracellular vesicles promotes endothelial cell migration via the ERK1/2 and JNK signaling pathways	Promote metastasis	[78]
Colorectal cancer	miR-27b-3p	LOVO, HCT-116, DLD-1, SW620 and SW480	Promote circulating tumor cell-mediated metastasis by modulating vascular permeability in colorectal cancer	Promote metastasis	[79]
Colorectal cancer	lncRNA-APC1	HCT-116	Enhance tumor angiogenesis by activating the MAPK pathway in endothelial cells.	Promote metastasis	[80]

**Abbreviations:** CRLM liver metastasis of colorectal cancer, MAPK mitogen-activated protein kinase, ANGPTL1 angiopoietin-like protein 1, MMP9 matrix metallopeptidase 9.

## Data Availability

Not applicable.

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
