# Peer review of "The Biological Effect of Small Extracellular Vesicles on Colorectal Cancer Metastasis"

_cells, 2022, doi:10.3390/cells11244071_

Round 1
Reviewer 1 Report
In this manuscript, Wang et al. present a summary of the biological effect of small extracellular vesicles in colorectal cancer metastasis. In this review, the authors described that sEVs mediate epithelial mesenchymal transition (EMT), reconfigured the tumor microenvironment (TME), modulated the immune system, and altered vascular permeability and angiogenesis to promote CRC metastasis, also discussed the current difficulties in studying sEVs and propose new ideas. While the manuscript is well organized, some other areas of the manuscript require clarification before publication. The manuscript is interesting and well organized, I think it can be accepted for the publication in this journal.
Author Response
We appreciate the reviewer’s positive evaluation of our work ,and thanks very much for taking your time to review this manuscript.
Reviewer 2 Report
The review seems very comprehensive. However, there are other recently published reviews in the field of SEVs and cancer. I think it would be very helpful if the authors could cite some of the recently published reviews in the domain of SEVs and cancer, and highlight the novelty of the current review. This would better help the authors to reach their target audience.
The English language used in the manuscript can be improved overall to improve the accessibility of the paper to a worldwide audience. I suggest you have a colleague who is proficient in English and familiar with the subject matter, review your manuscript, or contact a professional editing service.
Author Response
We are very grateful to your comments for the manuscript. According to your advice, we amended the relevant part in manuscript. All of your questions were answered one by one.
Question 1: The review seems very comprehensive. However, there are other recently published reviews in the field of SEVs and cancer. I think it would be very helpful if the authors could cite some of the recently published reviews in the domain of SEVs and cancer, and highlight the novelty of the current review. This would better help the authors to reach their target audience.
Response: We are grateful for the suggestion. To be more clear and in accordance with the reviewer concerns, we have added a brief description as follows: For the function of sEVs in promoting cancer metastasis through immunomodulation, previous researchers have mainly focused on cancer-derived sEVs that exert tumor host immunosuppressive functions and promote tumor progression by binding to and altering the biological functions of surface receptors on natural killer cells, dendritic cells, T and B lymphocytes, and mast cells. In recent years, more and more studies have focused on the mechanism by which sEVs mechanisms that induce differentiated macrophages to M2-type tumor-associated macrophages, providing new evidence for immunotherapy of cancer. We summarize the mechanistic study of how sEVs in CRC induce macrophages to undergo M2 polarization and thus disrupt anti-tumor immunity and enable CRC to metastasize, and how the use of sEVs to inhibit macrophages from undergoing M2 polarization reduces metastasis in colorectal cancer, making it a How to reduce the metastasis of colorectal cancer by using sEVs to inhibit M2 polarization of macrophages and make it a promising new target for tumor treatment will be the focus of our future research.
Question 2: The English language used in the manuscript can be improved overall to improve the accessibility of the paper to a worldwide audience. I suggest you have a colleague who is proficient in English and familiar with the subject matter, review your manuscript, or contact a professional editing service.
Response: We apologize for the language problems in the original manuscript. The language presentation was improved by contacting a professional editing service.
Reviewer 3 Report
The manuscript is documented well, but I have some questions which must be addressed.
The manuscript states that small extracellular vesicles play a significant role in the metastasis s of con cancer. The author did not mention the colonization of metastatic cells colonization at the Secondary sited. EMT plays an important role in metastasis. Tight junction proteins are essential proteins that maintain the integrity of cells' structure and selective permeability and play a crucial role in maintaining extra-cellular vesicle stability and regulating budding (PMID 15269339). A Mountain of evidence suggests that tight junction protein plays a vital role in cell polarization and differentiation. For example, claudin3 loss-induced EMT (PMID: 28783170) and loss of claudin-2 loss in renal cancer promote EMT (PMID: 33622361).
Author Response
Thank you for your summary. We really appreciate your efforts in reviewing our manuscript. We have revised the manuscript accordingly. Our point-by-point responses are detailed below.
Question 1: The manuscript states that small extracellular vesicles play a significant role in the metastasis s of con cancer. The author did not mention the colonization of metastatic cells colonization at the Secondary sited.
Response:Thank you for your precious comments and advice. Those comments are all valuable and very helpful for revising and improving our paper. In our review it was suggested that secondary colonization of tumor cells is determined by the local microenvironment of distant organs. Primary tumors are known to contribute to the formation of a favorable microenvironment at secondary sites, also known as pre-metastatic ecological niches. Therefore, we decided to discuss only small extracellular vesicles promoting metastasis in colorectal cancer by influencing the pre-metastatic ecological niche of tumors. We mentioned the above in the Small extracellular vesicles affect colorectal cancer metastasis by remodeling the tumor microenvironment section of the article.
Question 2: EMT plays an important role in metastasis. Tight junction proteins are essential proteins that maintain the integrity of cells' structure and selective permeability and play a crucial role in maintaining extra-cellular vesicle stability and regulating budding (PMID 15269339). A Mountain of evidence suggests that tight junction protein plays a vital role in cell polarization and differentiation. For example, claudin3 loss-induced EMT (PMID: 28783170) and loss of claudin-2 loss in renal cancer promote EMT (PMID: 33622361).
Response:Thank you for your suggestion. As suggested by reviewer, we have added the suggested content to the manuscript on page 6 as follows: In addition, tight junction proteins are responsible for regulating para-cellular permeability and maintaining cell polarity(PMID 15269339). The absence of tight junction proteins can lead to the development of EMT and further promote cancer progression(PMID 33622361, PMID 28783170). Evidence suggests that the expression of Claudin-2 in small extracellular vesicles of patient blood origin can be a relevant prognostic biomarker for predicting the development of replacement type liver metastases in patients with colorectal cancer. replacement type liver metastases(PMID 34079064).
Round 2
Reviewer 3 Report
Excellent study. I would like to publish this research article.